# A Comparison of Rice Root Microbial Dynamics in Organic and Conventional Paddy Fields

**DOI:** 10.3390/microorganisms13010041

**Published:** 2024-12-29

**Authors:** Fangming Zhu, Takehiro Kamiya, Toru Fujiwara, Masayoshi Hashimoto, Siyu Gong, Jindong Wu, Hiromi Nakanishi, Masaru Fujimoto

**Affiliations:** 1Department of Agricultural and Environmental Biology, Graduate School of Agricultural and Life Sciences, The University of Tokyo, Tokyo 113-8657, Japan; zhufangming96@g.ecc.u-tokyo.ac.jp (F.Z.);; 2Department of Applied Biological Chemistry, Graduate School of Agricultural and Life Sciences, The University of Tokyo, Tokyo 113-8657, Japan; 3Department of Global Agricultural Sciences, Graduate School of Agricultural and Life Sciences, The University of Tokyo, Tokyo 113-8657, Japan

**Keywords:** rice, root-associated microbiome, developmental stage, organic and conventional paddies, amplicon sequencing, diversity analysis

## Abstract

The assembly of plant root microbiomes is a dynamic process. Understanding the roles of root-associated microbiomes in rice development requires dissecting their assembly throughout the rice life cycle under diverse environments and exploring correlations with soil properties and rice physiology. In this study, we performed amplicon sequencing targeting fungal ITS and the bacterial 16S rRNA gene to characterize and compare bacterial and fungal community dynamics of the rice root endosphere and soil in organic and conventional paddy fields. Our analysis revealed that root microbial diversity and composition was significantly influenced by agricultural practices and rice developmental stages (*p* < 0.05). The root microbiome in the organic paddy field showed greater temporal variability, with typical methane-oxidizing bacteria accumulating during the tillering stage and the amount of symbiotic nitrogen-fixing bacteria increasing dramatically at the early ripening stage. Redundancy analysis identified ammonium nitrogen, iron, and soil organic matter as key drivers of microbial composition. Furthermore, correlation analysis between developmental stage-enriched bacterial biomarkers in rice roots and leaf mineral nutrients showed that highly mobile macronutrient concentrations positively correlated with early-stage biomarkers and negatively correlated with later-stage biomarkers in both paddy fields. Notably, later-stage biomarkers in the conventional paddy field tended to show stronger correlations with low-mobility nutrients. These findings suggest potential strategies for optimizing microbiome management to enhance productivity and sustainability.

## 1. Introduction

Rice is one of the world’s most important staple crops, providing dietary energy for more than 50% of the global population [1,2]. As the global population continues to grow, ensuring stable rice production despite the constraints such as limited, degraded arable land and climate change has become a critical challenge in agriculture [3]. Farmers primarily rely on chemical fertilizers and pesticides to increase crop yields [4,5]. However, long-term dependence on chemical inputs has caused environmental issues, such as soil acidification, groundwater contamination, and greenhouse gas emissions [6,7]. Therefore, developing greener, more sustainable agricultural practices is crucial for the future of agriculture.

In recent years, the role of microorganisms in agroecosystems has attracted significant attention [8,9]. The development and the utilization of beneficial microbiomes have been proposed as a sustainable alternative to chemical inputs [10,11]. Soil microbes are essential for organic matter decomposition, nutrient cycling, and pathogen control, and they also promote plant health through interactions with roots [12,13,14]. To effectively manage these beneficial microbial communities, it is necessary to understand how agricultural management affects the selection of root-associated communities. It has been demonstrated that agricultural management influenced microbial community composition in roots and soils [15,16,17]. Long-term organic management has been shown to maintain crop yields that are comparable to those of conventional agriculture while significantly increasing microbial abundance and bacterial diversity in soil ecosystems [18,19,20]. In contrast, conventional systems, characterized by the widespread application of chemical fertilizers and pesticides, tend to reduce soil microbial diversity and community heterogeneity, potentially weakening the resilience and resistance of soil microbiomes [21,22]. Despite these insights, our understanding of the differences in root-associated and soil microbial communities between organic and conventional paddies remains limited, particularly concerning the dynamics of microbial diversity and ecological function under these contrasting management practices.

Plant roots adapt to their environment by recruiting and assembling beneficial microorganisms from the surrounding soil, collectively known as plant growth-promoting microorganisms (PGPMs) [23]. These endophytes have a significant impact on host plant health and the productivity of host plants by enhancing nutrient uptake, regulating growth, and strengthening resistance to pathogens and abiotic stresses [24,25,26]. The assembly of microbial communities around rice roots is a dynamic process influenced by the soil type, developmental stage, and genotype [27,28,29,30]. Previous studies highlighted how changes in root exudate composition throughout plant development shape rhizosphere microbial communities over time [8,31]. These microbes play essential roles in promoting nutrient uptake, enhancing growth, improving tolerance to abiotic stresses, and providing resistance to pathogens [12,13,14]. Therefore, understanding the microbial composition and drivers of rice root-associated microbial communities is vital for harnessing beneficial microbes in modern agriculture.

We hypothesized that agricultural management, particularly farming methods, and rice development influence the composition and function of root-associated microbial communities across developmental stages by modulating host and environmental factors. To test this, we performed 16S rRNA and internal transcribed spacer (ITS) gene amplicon sequencing on the rice variety ‘Koshihikari’, widely cultivated in Japan, under organic and conventional paddy field conditions, which differed in fertilizer type and the use of agrochemicals, including herbicides, insecticides, and fungicides. Specifically, this study aimed to (1) compare the composition of rice root-associated microbial communities under organic and conventional paddy field conditions across different developmental stages to capture temporal dynamics; (2) identify microbial biomarkers specific to each paddy management system and developmental stage; and (3) evaluate correlations between key microbial taxa and soil physicochemical properties, as well as rice leaf mineral nutrient profiles. Based on these findings, we provide significant theoretical insights into optimizing sustainable rice production systems and develop strategies to harness microbial resources for enhancing productivity and environmental sustainability.

## 2. Materials and Methods

### 2.1. Experimental Design

This experiment was conducted in Ryugasaki City, Ibaraki, Japan (35.9° N, 140.2° E), across two distinct paddy fields approximately 300 m apart: one managed with an organic farming practice, measuring approximately 6 × 87 m, and the other managed with a conventional farming practice, measuring approximately 29 × 92 m. In 2021, the Japanese major rice cultivar Koshihikari (*Oryza sativa* L.) was sown on 4 April in organic practice and on 10 April in conventional practice. Rice seedlings were transplanted into the organic paddy field on 15 May with an organic fertilizer containing 4.8 kg N, 3.5 kg P, and 1.7 kg K per 1000 m^2^, and into the conventional paddy field on 3 May with a chemical fertilizer containing 6 kg N, 6 kg P, and 6 kg K per 1000 m^2^. An additional organic fertilizer containing 2.1 kg N, 1.2 kg P, and 0.7 kg K per 1000 m^2^ was applied to the organic paddy field on 5 July. In the organic paddy field, weed control was achieved mechanically, and pests and diseases were managed manually, whereas in the conventional paddy field, these tasks were handled using agricultural chemicals, including the herbicide pretilachlor, the insecticide dinotefuran, and the microbicide azoxystrobin. Whole rice plants were sampled at four developmental stages: tillering (10 June), elongating (9 July), early ripening (4 August), and maturing (25 August). At each rice developmental stage, one rice plant was sampled from each of three distinct plots (each approximately 2 × 20 m in size) in both fields (n = 3 per paddy field per stage). In total, 24 rice plants were sampled across four developmental stages from the organic and conventional paddy fields. Whole rice plants, along with the surrounding soil, were collected with a clean shovel and transported to the lab for processing. Roots were excised 5–10 cm from the base with sterile scissors, washed three times in 20 mL of 1 × PBS within a 50 mL centrifuge tube, and dried with sterile paper to remove residual moisture. These roots were then chopped into 2 mm fragments and placed in Lysing Matrix E tubes (MP biomedicals, Irvine, CA, USA) for DNA extraction. Sterile spoons were used to collect about 300 mg of bulk soil from a depth of 5–10 cm below the soil surface. Each soil sample was placed into Lysing Matrix E tubes for DNA extraction. The remaining soil was prepared for physicochemical property analysis.

### 2.2. Determination of Soil Physicochemical Properties

Soil samples were air-dried, mixed thoroughly, and passed through a 2 mm sieve. A minimum of 100 g of each sieved sample were analyzed for physicochemical properties by Katakura Co-op Agri Corporation, Tsukuba Analysis Center (Tsuchiura City, Ibaraki, Japan). The detailed analytical methods are shown in Appendix A.

### 2.3. Determination of Shoot Dry Weight and Panicle Dry Weight

Rice plants were harvested at the maturing stage to measure shoot and panicle dry weight. Shoots and panicles were manually separated, rinsed thoroughly with deionized water to remove adhering soil and dust, then oven-dried at 70 °C for 72 h until a constant weight was achieved. The dry weights were recorded using a precision analytical balance. All measurements were performed in triplicate to ensure reproducibility.

### 2.4. DNA Extraction, PCR Amplification and Sequencing

Following the manufacturer’s instructions, microbial DNA was extracted from root and soil samples using the SPINeasy DNA Kit for Soil (MP Biomedicals, Irvine, CA, USA). The DNA concentration and purity were assessed by a NanoDrop One C Spectrophotometer (Thermo Fisher Scientific, Madison, WI, USA). The variable V3–V4 region of the bacterial 16S rRNA gene and the fungal ITS1 region were amplified from each sample using primer pairs 338F (5′-ACTCCTACGGGAGGCAGCAG-3′)/806R (5′-GGACTACHVGGGTWTCTAAT-3′) for bacteria and ITS-1F (5′-CTTGGTCATTTAGAGGAAGTAA-3′)/ITS-2R (5′-GCTGCGTTCTTCATCGATGC-3′) for fungi. PCR enrichment was performed in a 50 μL reaction volume containing 30 ng of template, fusion PCR primers, and PCR master mix. The following PCR amplification program was used: pre-denaturation at 95 °C for 3 min; denaturation at 95 °C for 15 s, annealing at 56 °C for 15 s, extension at 72 °C for 45 s, a total of 30 cycles; final extension at 72 °C for 5 min. The amplified products were purified by Agencourt AMPure XP beads (Beckman Coulter, Indianapolis, IN, USA), then eluted in Elution Buffer and subsequently labeled for library preparation. The fragment size and concentration of the constructed libraries were assessed using the Agilent 2100 Bioanalyzer (Agilent Technologies, Santa Clara, CA, USA). Sequencing was conducted on the Illumina MiSeq platform (Illumina, BGI-Shenzhen, Shenzhen, China) with a paired-end protocol. Sequence data were deposited into the NCBI Sequence Read Archive (SRA) database with the accession numbers PRJNA1177627 and PRJNA1180211.

### 2.5. Measurement of Leaf Mineral Nutrient Concentrations

The uppermost fully expanded leaves of rice were collected at each developmental stage and dried for mineral nutrient analysis. The homogenized samples were treated with 61% (*v*/*v*) HNO_3_ and dissolved in 0.08 M of HNO_3_ containing 1 ppb of indium as an internal standard. Mineral nutrient concentrations were measured by inductively coupled plasma mass spectrometry (ICP-MS; Agilent 7800, Agilent Technology, Santa Clara, CA, USA). Carbon (C) and nitrogen (N) concentrations were calculated using a CN corder (vario MAX cube; Elementar, Yokohama, Japan).

### 2.6. Microbiome Data Analysis

Microbiome bioinformatics analysis was performed using QIIME 2 (version 2023.5) [32]. Raw paired-end sequence data generated on the Illumina MiSeq platform were imported into the QIIME 2 platform using the qiime tools import command. The sequences were demultiplexed based on barcode information to generate demultiplexed sequence files (qiime demux). Low-quality sequences, sequencing noise, and chimeric sequences were filtered out using DADA2 (qiime dada2 denoise-paired) [33], resulting in representative sequences and a table of amplicon sequence variants (ASVs), also referred to as the feature table. Representative sequences were aligned against reference databases for taxonomic annotation: the SILVA 138 database for 16S rRNA sequences and the UNITE database (version 9.0) for ITS1 sequences [34,35], using the qiime feature-classifier classify-sklearn command. Sequences identified as “Chloroplast” or “Mitochondrial” were removed to exclude plant- and host-derived contaminants using the qiime taxa filter-table command. The resulting taxonomy table assigned each feature to its corresponding taxonomic level (e.g., phylum, class, order, family, genus, and species). The taxonomy table was used for subsequent analyses, including differential abundance analysis, microbial community composition analysis, and correlation analysis. Differentially abundant microbiota between groups were identified using the linear discriminant analysis (LDA) effect size (LEfSe) with an LDA threshold of 2 and a *p*-value threshold of 0.05. LEfSe analysis was performed on the Huttenhower Lab Galaxy server (http://huttenhower.sph.harvard.edu/galaxy (accessed on 19 February 2024). Microbial diversity was estimated based on the feature table. Sequences were aligned using MAFFT (qiime alignment mafft, version 7.520) [36] and the phylogenetic tree was constructed with FastTree (qiime phylogeny fasttree, version 2.1.11) [37]. Alpha and beta diversity metrics were calculated using the QIIME 2 q2-diversity plugin (version 2023.5).

### 2.7. Plotting and Statistical Analysis

Boxplots were generated using the ggplot2 package in R software (version 4.3.2). Principal coordinates analysis (PCoA) based on Bray–Curtis dissimilarities was also performed using the ggplot2 package in R [38], and the variance explained by the first two principal coordinates (PCoA1 and PCoA2) was displayed on the respective axes. The VennDiagram package was adopted to display the distribution of unique and shared genera among different samples. The bar plots of microbial composition were generated by Microsoft Excel for Mac (version 16.90.2). A heat map of the relative abundance of microbial taxa was constructed in TB Tools (version 2.127) [39]. Clustering correlation heatmapping with signs was performed using the OmicStudio tools (https://www.omicstudio.cn (accessed on 15 October 2024).

Significant differences among groups were determined using one-way analysis of variance (ANOVA), followed by Tukey’s Honest Significant Difference (HSD) test for post hoc comparisons. Statistical significance levels were defined as follows: *p* < 0.05 (*), *p* < 0.01 (**), and *p* < 0.001 (***). These symbols were used to indicate the degree of significance in figures and tables. Cohen’s *d* was used to supplement ANOVA to assess the practical significance of differences between organic and conventional paddies. Large effect sizes (*d* > 0.8) were reported and highlighted, even when no statistical significance (*p* > 0.05) was detected, to capture meaningful ecological differences between the two paddy types. Permutational multivariate analysis of variance (PERMANOVA) was conducted to evaluate the effects of compartment, paddy type, and developmental stage on the microbial community composition. The analysis was performed using the adonis2 function in the R package vegan (v2.6-8) with 999 permutations. To assess the significance of environmental variables in explaining dominant microbial composition, redundancy analysis (RDA) was performed using Canoco 5 (version 5.15) [40]. The Spearman correlation coefficient method was used to analyze the correlation between bacterial biomarkers’ relative abundance and leaf mineral nutrient concentrations. All analyses were conducted using R software (version 4.3.2). All Figures were assembled using Adobe Illustrator 2025 software (version 29.0.1).

## 3. Results

### 3.1. Microbial Diversity and Richness Respond to Rice Development and Agricultural Practices

In our 16S rRNA and ITS1 amplicon sequencing, a total of 9823 bacterial ASVs and 2274 fungal ASVs were obtained from the 2,372,445 and 668,841 quality-filtered sequences, respectively. Based on the relative abundance and counts of each ASV, we characterized the diversity and richness of microbial communities using the Shannon index and observed features (Figure 1).

The Shannon index of bacterial communities was significantly influenced by compartment (*p* < 0.001), paddy type (*p* < 0.05), and developmental stage (*p* < 0.001; Appendix A). Bacterial diversity in the root endosphere was consistently lower than that in soil across all developmental stages (*p* < 0.05; Figure 1A). During the tillering stage, the organic paddy exhibited significantly lower bacterial diversity in the root endosphere compared to the conventional paddy, while no significant differences were observed in soil compartments across stages. Significant two-way (Compartment × Paddy) and three-way (Compartment × Paddy × Stage) interactions for the Shannon index (*p* < 0.05; Appendix A) further highlighted the complex interdependencies between paddy type, compartment, and developmental stage in shaping bacterial diversity. Observed features of bacterial communities were significantly influenced by compartment (*p* < 0.001) and developmental stage (*p* < 0.001; Appendix A), while no significant effects were observed for paddy type (*p* > 0.05). However, substantial differences, indicated by large effect sizes (Cohen’s *d* > 0.8), were detected during the tillering and maturing stages in the root endosphere (Figure 1B). These findings suggest that organic management practices may influence rice root microbial diversity and functionality, particularly during early and late developmental stages.

The Shannon index and observed features of fungal communities were significantly influenced by compartment (*p* < 0.001) and paddy type (*p* < 0.05), while developmental stage had no significant effect (*p* > 0.05; Appendix A). Fungal diversity and observed features in the root endosphere were consistently lower than those in soil (*p* < 0.05; Figure 1C,D). Notably, although no statistical significance was detected between organic and conventional paddies, large effect sizes (Cohen’s *d* > 0.8) were observed during specific stages, particularly the elongating and early ripening stages, where the root endosphere in the organic paddy exhibited higher fungal diversity and richness (Figure 1C,D).

The compartment consistently emerged as the dominant factor shaping microbial diversity and feature richness across all analyses (*p* < 0.001; Appendix A). Root-associated microbial communities exhibited significantly lower diversity and feature richness compared to soil, reflecting the distinct ecological constraints and selection pressures present in the root microenvironment. These findings highlight the critical role of the compartment and the influence of organic management practices in shaping microbial community structure and functionality, particularly within root-associated niches.

### 3.2. Microbial Communities Are Predominantly Shaped by Compartments, Agricultural Practices, and Developmental Stages

To assess variations in microbial composition across different groups, we calculated Bray–Curtis distances based on the microbial feature table and visualized beta diversity using principal coordinate analysis (PCoA) (Figure 2). For all groups, microbial communities in bulk soil and the root endosphere formed distinct clusters, with PCoA1 explaining 35.96% of the variation in bacterial communities (Figure 2A) and 12.65% in fungal communities (Figure 2B). PERMANOVA results for both bacteria and fungi supported the PCoA results, showing that the majority of the variation in microbial communities was attributed to the sample compartment (35.5% of variance in bacteria; *p* = 0.001; 55.6% of variance in fungi; *p* = 0.001).

To further investigate the impact of agricultural management practices on microbial community composition across four developmental stages (tillering, elongating, early ripening, and maturing), PCoAs were performed separately on the root endosphere and bulk soil under organic and conventional paddy conditions. The results revealed that both bacterial and fungal communities in the root endosphere (Figure 2C,D) and bulk soil (Figure 2E,F) clustered distinctly into two groups according to paddy type. PCoA1 (25.36% in Figure 2C and 27.65% in Figure 2E) for bacteria and PCoA1 (17.74% in Figure 2D and 22.25% in Figure 2F) for fungi explained the majority of the variation in the root endosphere and bulk soil, respectively. These observations were consistent with the PERMANOVA results, which demonstrated significant differences in microbial community composition between conventional and organic paddy fields (*p* < 0.05). These findings suggest that agricultural management practices exert a strong influence on the structure of microbial communities in both the root endosphere and bulk soil.

Additionally, with the exception of fungal communities in bulk soil (5.4% of variance, *p* = 0.142; Figure 2F), the developmental stage also affected the structure of microbial communities, including bacterial communities in the root endosphere (16.5% of variance, *p* = 0.001; Figure 2C) and bulk soil (7.9% of variance, *p* = 0.044; Figure 2E), as well as fungal communities in the root endosphere (10.4% of variance, *p* = 0.001; Figure 2D). These results suggest that the root may harbor more dynamic shifts in microbial communities across developmental stages in response to rice growth.

### 3.3. Root Microbial Structure Changes Across Rice Developmental Stages

The taxonomic composition analysis of rice root endosphere and bulk soil microbial communities revealed distinct patterns of microbial dynamics across four developmental stages (Figure 3). The number of genera in bacterial and fungal communities in the root endosphere was smaller than that in bulk soil (*p* < 0.01 in bacteria, *p* < 0.001 in fungi; Appendix A). Nevertheless, the taxonomic composition of bacterial and fungal communities in the root endosphere (Figure 3A,C) tended to exhibit greater temporal shifts compared to those in bulk soil (Figure 3B,D), which displayed greater stability. In the taxonomic composition analysis of bacterial communities, *Alphaproteobacteria*, *Myxococcota*, and *Gammaproteobacteria* dominated the root endosphere (Figure 3A), while *Gammaproteobacteria*, *Acidobacteriota*, and *Chloroflexi* were predominant in the bulk soil (Figure 3B). Notably, the relative abundance of *Alphaproteobacteria* in the root endosphere increased significantly with the progression of rice developmental stages (*p* < 0.05; Figure 3A and Appendix A).

The fungal communities in the root endosphere (Figure 3C) and bulk soil (Figure 3D) were dominated by *Ascomycota* and *Basidiomycota*, aligning with findings from previous studies on paddy soils [41,42]. Notably, in the organic paddy field, a gradual domination of *Fungi_phy_Incertae_sedis* in the root endosphere (Figure 3C) and a temporal increase in *Glomeromycota*, known for forming arbuscular mycorrhiza, in the bulk soil (Figure 3D) were observed. This result indicates that functional shifts in fungal communities of rice roots and soil in the organic paddy field, including symbiotic interactions with rice roots, occur along with rice developmental stages.

In the comparative analysis of the microbial taxonomic composition between the two paddy field types, bacterial communities in the root endosphere under the organic paddy field exhibited a larger enrichment in *Gammaproteobacteria* across rice developmental stages (Appendix A) and a temporal enrichment of *Verrucomicrobiota* and *Chloroflexi* during the elongating and early ripening stages (Figure 3A). In contrast, under the conventional paddy, the taxonomic compositions of bacterial communities (Figure 3B) and fungal communities (Figure 3D) in bulk soil were more stable across developmental stages than those under the organic paddy. These results suggest that organic management practices may lead to higher temporal variability in bacterial and fungal community dynamics.

### 3.4. Distinct Core Microbiomes Exist in Root and Soil

To explore microbial dynamics and the influence of management practices on rice microbial structure, we further analyzed the unique and shared genera across four developmental stages to identify core taxa that are essential for the rice growth in bacterial communities (Figure 4) and fungal communities (Appendix A). In the root endosphere, 106 shared bacterial genera (Figure 4A) and 14 shared fungal genera (Appendix A) were identified in the organic paddy, while 138 shared bacterial genera (Figure 4B) and 7 shared fungal genera (Appendix A) were found in the conventional paddy. Taxonomic analysis of the core root bacterial microbiome across all stages revealed the dominance of key phyla, including *Gammaproteobacteria* (18.8%), *Alphaproteobacteria* (11.6%), *Acidobacteriota* (10.1%), *Chloroflexi* (10.1%), *Bacteroidota* (10.1%), and *Spirochaetota* (10.1%) (Figure 4E). Meanwhile, unique genera at different stages in these analyses are considered to highlight the temporal dynamics of bacterial and fungal communities associated with rice growth.

In bulk soil, 354 bacterial (Figure 4C) and 41 fungal (Appendix A) genera were shared across stages in the organic paddy, compared to 336 bacterial (Figure 4D) and 36 fungal genera (Appendix A) in the conventional paddy. Dominant phyla included *Gammaproteobacteria* (16.7%), *Acidobacteriota* (16.1%), *Alphaproteobacteria* (12.9%), *Actinobacteriota* (10.0%), and *Bacteroidota* (12.4%), which play essential roles in nutrient cycling, organic matter decomposition, and soil health maintenance (Figure 4F) [43,44,45,46]. These results suggest that the consistent presence of these bacterial phyla across developmental stages in both organic and conventional paddies highlights their important roles in maintaining root and soil health, as well as promoting rice growth.

### 3.5. Soil Physicochemical Factors Drive Rice Root-Associated Microbial Phyla

To identify potential environmental drivers influencing the characteristics of rice root and soil microbial communities, we performed redundancy analysis (RDA) to explore the relationships between dominant microbial communities and soil physicochemical properties (Figure 5). For bacterial communities, RDA1 and RDA2 accounted for 35.42% and 22.61% of the variation in the root endosphere (Figure 5A) and 47.29% and 20.09% in the bulk soil (Figure 5B), respectively. In the root endosphere, ammonium nitrogen (NH_4_^+^-N) and iron (Fe) have stronger effects compared to other environmental factors on the composition of the bacterial community, showing significant positive correlations with *Gammaproteobacteria* and *Spirochaetota* (*p* < 0.01) (Figure 5A). In bulk soil, Fe and available silica (AS) were significantly correlated with *Actinobacteriota* (*p* < 0.01) (Figure 5B). pH (*p* < 0.05) and NH_4_^+^-N (*p* < 0.01) were also notable contributors to bacterial variability, positively impacting *Actinobacteriota*, *Gammaproteobacteria*, *Alphaproteobacteria*, and *Bacteroidota* (Figure 5B), implying that bacterial community structure in bulk soil is significantly influenced by soil physicochemical properties. The distinct influence of nitrogen forms, especially NH_4_^+^, on the characteristics of the soil bacterial community suggests the importance of nitrogen cycling processes, particularly in the assembly of *Gammaproteobacteria*, *Alphaproteobacteria*, and *Bacteroidota* within paddy soil environments.

In the RDA results for fungal communities, a total of 54.61% of the variance (RDA1: 36.48%, RDA2: 18.13%) in the root endosphere (Figure 5C) and 28.61% of the variance (RDA1: 19.99%, RDA2: 8.62%) in bulk soil (Figure 5D) were explained by soil physicochemical factors. In the root endosphere, factors such as NH₄⁺-N and soil organic matter (SOM) were identified as the primary factors influencing the distribution of dominant taxa, including *Ascomycota* and *Basidiomycota* (Figure 5C). Additionally, SOM exhibited a significant negative correlation with *Ascomycota* in the bulk soil (*p* < 0.01; Figure 5D). Notably, Magnesium (Mg) and potassium (K) appeared to influence the properties of fungal communities, with distinct responses observed among fungal taxa in both the root endosphere (Figure 5C) and bulk soil (Figure 5D). Across both bacterial and fungal communities, these results demonstrated that soil physicochemical factors, such as ammonium nitrogen and iron, and soil organic matter are key drivers of microbial diversity and composition.

### 3.6. Distinct Root and Soil Biomarkers Characterize Organic and Conventional Paddies

To better understand the effects of different management practices on rice root microbial communities, we identified genus-level microbial biomarkers distinguishing between organic and conventional paddies based on linear discriminant analysis (LDA). This analysis revealed significant differences in bacterial and fungal biomarkers between the two paddy types at each developmental stage of rice (Figure 6). For bacterial communities, in the root endosphere, bacterial taxa such as *Uliginosibacterium* and *Novosphingobium* were significantly enriched in the organic paddy (Figure 6A), particularly during the reproductive stages (elongating, the early ripening, and maturing stages) (Figure 6E). In contrast, bacterial taxa such as *Anaeromyxobacter*, *Leptonema*, *Sporomusaceae_uncultured*, and *Geobacteraceae*, were enriched in the conventional paddy (Figure 6A) during the tillering, elongating, or early ripening stages, though their relative abundance declined by the maturing stage (Figure 6E). In the bulk soil, most bacterial biomarkers in the conventional paddy, as shown in Figure 6B, exhibited a similar trend, with higher relative abundances of *RBG_13_54_9*, *Gaiella*, and *MB_A2_108* during pre-flowering stages (tillering and elongating stages), followed by a marked decline during the post-flowering stages (early ripening and maturing stages) (Figure 6F). Conversely, among the soil bacterial biomarkers for the organic paddy, as shown in Figure 6B, bacterial taxa such as *SC_I_84* and *Acidobacteriota_subgroup_7* were enriched during the pre-flowering stages, whereas *Candidatus_Nitrosotalea* and *MND1* significantly accumulated in the post-flowering stages (Figure 6F).

For the fungal community, in the root endosphere of the organic paddy, greater enrichment of taxa such as *Fungi_gen_Incertae_sedis*, *Ascomycota_unclassified*, *Thanatephorus*, and *Branch06_gen_Incertae_sedis* was observed (Figure 6C). Notably, fewer fungal biomarkers were identified in the root endosphere of the conventional paddy (Figure 6C), with only taxa *Ascomycota_gen_Incertae_sedis* and *Pezizaceaes_gen_Incertae_sedis* dominating during the elongating and early ripening stages (Figure 6G). In the bulk soil, *Junewangiaceae_gen_Incertae_sedis*, *Staphylotrichum*, *Branch06_gen_Incertae_sedis*, and *Mortierella* were specifically enriched in the organic paddy (Figure 6D), while fungal taxa such as *Trichoderma*, *Podospora*, and *Neurospora* were enriched in the conventional paddy, exhibiting distinct community compositions across different developmental stages (Figure 6H). These findings suggest that bacterial and fungal biomarkers specific to rice roots and soil differ between organic and conventional paddies, and that these biomarkers are enriched at distinct stages throughout the entire rice growth cycle.

### 3.7. Stage-Specific Root Biomarkers Are Enriched Across Developmental Stages

Considering the stage-specific enrichment of some biomarkers described above, we identified key biomarkers associated with each paddy type at different developmental stages for root bacterial communities (Figure 7), soil bacterial communities (Appendix A), and root and soil fungal communities (Appendix A). In the organic paddy, the root endosphere was enriched with bacterial taxa such as *Comamonadaceae*, *Methylomonas*, *Bradyrhizobium*, *Azospirillum,* and *Burkholderia_Caballeronia_Paraburkholderia* (Figure 7A). These taxa are well known for their association with beneficial soil processes, including nitrogen fixation, methane oxidation, and plant growth promotion [47,48,49,50]. In particular, *Comamonadaceae* and *Methylomonas*, which were enriched during the tillering stage, play important roles in methane metabolism and nitrogen cycling [51,52]. Additionally, typical symbiotic nitrogen-fixing genera such as *Bradyrhizobium* and *Azospirillum* were highly recruited during the early ripening stage in the organic paddy (Figure 7A). Similarly, the rice root recruited different functional bacterial taxa corresponding to different developmental stages in the conventional paddy (Figure 7B). Iron-oxidizing bacteria, such as *Sideroxydans* and *Rhodoferax*, which assist seedlings in acquiring essential mineral nutrients [53], were enriched at the tillering stage (Figure 7B). Meanwhile, key free-living nitrogen-fixing bacteria in paddy soils [54,55], such as *Anaeromyxobacter* and *Geobacteraceae* were particularly enriched in the conventional paddy (Figure 7B).

Following the identification of root-associated microbial biomarkers, we also screened bacterial biomarkers in bulk soil samples collected at different developmental stages (Appendix A). Although the functions of many bacterial biomarkers in bulk soil remained uncharacterized, bacterial genera such as *Massilia*, *Nitrospira*, *Gaiella*, *P9X2b3D02*, and *Nitrospirota_4_29_1* likely involved in nitrogen cycling [56,57,58,59] were detected at each stage in both organic and conventional paddy fields (Appendix A). Particularly in the conventional paddy, bacterial genera such as *Gaiella*, *Sphingomonas*, and *SBR1031*, likely involved in degradation of chemical compounds and microplastics [60,61,62,63], were identified as stage-enriched biomarkers for bulk soil (Appendix A).

However, compared to the bacterial communities described above, fungal communities exhibited fewer stage-enriched biomarkers in the rice root endosphere (Appendix A) and bulk soil (Appendix A). Nevertheless, the temporal enrichment of fungal genera likely associated with the plant pathogenicity or disease resistance, such as *Ustilaginoidea*, *Catenaria, Coniochaeta*, *Trichoderma*, and *Thanatephorus* [64,65,66,67,68], and those involved in soil organic matter degradation such as *Aspergillus*, *Coprinopsis*, *Cyberlindnera*, *Neurospora*, *Staphylotrichum*, and *Mucor* [69,70,71,72,73] was detected in both types of paddy fields (Appendix A). These findings reveal that the rice root and paddy soil harbor distinct microbial communities shaped by their respective management practices and stages of rice development.

### 3.8. Stage-Specific Enrichment of Root Biomarkers Correlates with Leaf Mineral Nutrient Dynamics

Next, to examine the correlations between stage-enriched root endosphere biomarkers and the mineral nutrient status of rice, we analyzed the correlations between these bacterial biomarkers and the concentrations of six macronutrients—nitrogen (N), phosphorus (P), potassium (K), sulfur (S), magnesium (Mg), and calcium (Ca)—and seven micronutrients—iron (Fe), manganese (Mn), zinc (Zn), copper (Cu), boron (B), molybdenum (Mo), and nickel (Ni)—in the fully expanded uppermost leaves of rice at four developmental stages in organic and conventional paddy fields (Figure 7C,D). In this analysis, bacterial biomarkers from the organic paddy exhibited stronger correlations with highly mobile nutrients such as N, P, K, S, Mg, and Zn (Figure 7C). Notably, at the tillering stage, these biomarkers showed significant positive correlations, while at the early-ripening and maturing stages, they exhibited significant negative correlations (*p* < 0.05; Figure 7C). Similarly to biomarkers from the organic paddy, those from the conventional paddy also showed significant positive correlations with highly mobile macronutrients at the tillering stage and negative correlations at the early-ripening and maturing stages ((*p* < 0.05; Figure 7D). However, unlike those from the organic paddy, biomarkers from the conventional paddy exhibited stronger correlations with low-mobility nutrients such as Ca, Fe, Mn, and Ni at the early-ripening and maturing stages (Figure 7D). Importantly, the concentrations of these highly mobile nutrients in rice leaves tended to peak at the tillering stage (Appendix A), when biomarkers from both types of paddy fields showed positive correlations with these nutrients. In contrast, the concentrations of low-mobility nutrients tended to be highest at the early-ripening or maturing stages (Appendix A), when biomarkers from the conventional paddy field showed positive correlations with them (Figure 7D). The results suggest that, as rice grows, the concentrations of highly mobile and low-mobility mineral nutrients in the leaves undergo distinct temporal shifts. Additionally, at each developmental stage, these mineral nutrient profiles tend to align with the correlation patterns with stage-enriched root biomarkers from different types of paddy fields.

## 4. Discussion

To develop sustainable strategies that harness microbial communities to improve crop yield and enhance soil resilience, it is crucial to understand how host plants (e.g., developmental stages), environmental conditions (e.g., soil properties), and agricultural practices (e.g., organic versus conventional systems) influence root-associated and soil microbial communities [74,75]. In this study, we conducted an in-depth analysis of microbial community assembly and dynamics within the rice root endosphere and paddy soil. This approach enabled us to reveal the significant regulatory effects of rice developmental stages and agricultural management practices on rice root-associated microbial communities.

### 4.1. Effects of Agricultural Practices on Microbial Diversity in Paddy Fields

Our study revealed significant effects of compartments, agricultural practices, and developmental stages on microbial diversity and community structure in rice paddies. Microbial diversity and feature richness in the root endosphere were consistently lower than those in bulk soil (*p* < 0.05; Figure 1), reflecting the unique ecological constraints and selection pressures of the root microenvironment [27,76]. This pattern may result from the dynamic regulation of carbon, oxygen, pH, and nutrient availability by plant roots [77,78]. Compartment emerged as the dominant factor influencing microbial diversity (*p* < 0.001; Appendix A), highlighting the critical role of root-associated niches in shaping microbial community composition. These findings align with previous studies suggesting that root-associated microbial communities are strongly influenced by host plants and their surrounding environment. This filtering effect indicates that roots serve as microbial niches, recruiting specific PGPMs to enhance nutrient uptake and stress resilience [10,79,80].

The organic paddy management significantly influenced microbial diversity, particularly in the root endosphere (Figure 1). During the tillering stage, bacterial diversity in the root endosphere was significantly lower in the organic paddy compared to the conventional paddy (*p* < 0.05; Figure 1A). However, as rice developed, bacterial diversity in the root endosphere increased significantly during the elongating stage, becoming comparable to that in the conventional paddy, potentially meeting the nutrient demands of early rice growth. This finding supports the hypothesis that organic paddies promote greater microbial fluctuations, likely due to increased soil carbon and nutrient content [81]. Microbial diversity is widely regarded as a key indicator of soil health and disease resilience [82]. While several studies have reported that organic systems tend to support higher microbial diversity compared to conventional systems [24,83,84,85], our findings only partially validate this conclusion. Except for significant differences during the tillering stage, no differences were observed across other developmental stages. In specific stages (e.g., tillering and maturing stages), the conventional paddy exhibited higher bacterial richness in the root endosphere, as indicated by large effect sizes (Cohen’s *d* > 0.8; Figure 1B). For fungal communities, the root endosphere in the organic paddy exhibited higher diversity and richness during the elongating and early ripening stages, although these differences were not statistically significant (*p* > 0.05; Figure 1C,D). Notably, large effect sizes (Cohen’s *d* > 0.8) were observed, indicating that the lack of statistical significance could be attributed to the relatively small sample size, which may have limited the statistical power to detect meaningful differences. Additionally, this discrepancy may be influenced by unique organic management practices, such as compost application without chemical microbicides [67], which likely promoted the growth of specific fungal groups (Appendix A) while potentially suppressing bacterial diversity and richness. Notably, despite the observed differences in microbial diversity and richness between organic and conventional paddies, these changes did not translate into significant differences in rice productivity. Although rice yield was not measured in this experiment, no significant differences in shoot and panicle weights at the maturing stage were observed between rice plants in the two paddy fields (Appendix A). These findings further support the feasibility of adopting organic cultivation as a sustainable alternative to conventional practices in rice production.

PCoA revealed that the sample compartment (root versus soil) (35.5% of variance in bacteria, *p* = 0.001; 55.6% of variance in fungi; *p* =0.001) was the primary driver of β-diversity in the paddy microbiome, with agricultural practices (7.6% of variance, *p* = 0.003; 8.8% of variance in fungi; *p* =0.001) and rice developmental stages (4.5% of variance, *p* = 0.039; 3.9% of variance in fungi; *p* =0.002) also affecting community composition (Figure 2). For both bacterial and fungal communities, agricultural practices exhibited a stronger influence on the microbial composition in the root endosphere and bulk soil, as evidenced by the greater separation along PCoA1 and significant PERMANOVA results (*p* < 0.05; Figure 2C–F). In the PCoA plot (Figure 2C), bacterial communities in rice roots from the conventional paddy clustered more closely, which might be explained by the prevalence of chemical input-dependent bacteria adapted to consistent nutrient availability (Figure 2C). In contrast, bacterial communities in the organic paddy were more dispersed in the PCoA plot (Figure 2C), suggesting greater variability within the community, potentially reflecting fluctuations associated with bacterial adaptation to different developmental stages under organic management.

Several studies have suggested that the long-term application of prescribed agricultural practices has profound effects on soil physicochemical properties, consequently altering the composition, structure, and function of microbial communities [86,87]. Consistent with these findings, our study also demonstrated that soil physicochemical properties substantially influenced microbial community assembly within the rice root endosphere and bulk soil, promoting the enrichment of specific microbial taxa. Notably, in the root endosphere bacterial community, NH₄⁺ and Fe concentrations in paddy soil were critical factors for the relative abundance of *Gammaproteobacteria* and *Spirochaetota* (Figure 5A). In the soil bacterial community, the relative abundance of *Actinobacteriota*, *Alphaproteobacteria*, *Bacteroidota*, and *Gammaproteobacteria* was significantly associated with NH₄⁺, pH, Fe, and available silicon in paddy soil (Figure 5B). For the fungal community, NH₄⁺ and SOM showed strong correlations with fungal taxa, underscoring their important roles in organic matter decomposition and nutrient cycling (Figure 5C,D). Therefore, the effective management of NH₄⁺ and Fe in paddy soil could be essential for harnessing the functional potential of microbial communities in both organic and conventional paddy fields during rice cultivation.

### 4.2. Microbial Composition and Roles in Rice Roots and Paddy Soil

In our taxonomic composition analysis (Figure 3), microbial communities in the rice root endosphere and bulk soil showed distinct compositions, with dominant taxa in each compartment reflecting their unique roles in the paddy ecosystem. For example, *Gammaproteobacteria*, *Myxococcota*, and *Alphaproteobacteria* were abundant in the root endosphere (Figure 3A), while *Acidobacteriota*, *Gammaproteobacteria*, and *Chloroflexi* were dominant in bulk soil (Figure 3B). These differences in dominant bacterial taxa between rice roots and paddy soil may result from the influence of root exudates, which promote the rapid growth of symbiotic microbes, and the nutrient-poor conditions of the bulk soil environment, which favor highly adaptive microorganisms [88].

Unlike the more stable composition of bulk soil (Figure 3B,D), the microbial composition in the root endosphere appeared to be more variable, shifting in response to the developmental stages of rice (Figure 3A,C). Our results showed that root microbial communities in both organic and conventional paddies change progressively with rice development. Notably, bacterial communities in the rice root endosphere displayed increased diversity and dynamic composition during the vegetative period (from tillering to elongating stages), which tended to stabilize during the reproductive period (post-elongating stages) and remained stable until rice maturity (Figure 3A). Additionally, *Proteobacteria*, *Bacteroidota*, and *Myxococcota* were highly abundant in the root endosphere, particularly in the organic paddy (Figure 3A), where beneficial microbial taxa were notably prevalent. Among these bacterial taxa, *Gamma-* and *Alphaproteobacteria*, known for promoting nutrient uptake and enhancing the disease resistance of plants, as well as *Actinobacteriota*, which exhibit significant inhibitory effects on plant pathogens, were included, with each peaking during different stages of rice development (Figure 3 and Appendix A). These findings suggest that organic management may support plant health by enhancing nutrient cycling and pathogen suppression [45,49,51].

Similarly to the bacterial communities described above, fungal communities also exhibited distinct compositional patterns between organic and conventional paddy fields. Although *Ascomycota* and *Basidiomycota* dominated fungal communities in both the root endosphere and bulk soil, the organic paddy displayed higher fungal compositional diversity across all developmental stages, particularly characterized by the presence of symbiotic fungal taxa such as *Mucoromycota* and *Glomeromycota* (Figure 3C,D). Specifically, *Mucoromycota* includes ectomycorrhizal symbionts, such as *Endogonomycetes_gen_Incertae_sedis*, which is identified as a stage-enriched fungal root biomarker (Appendix A). *Glomeromycota* function as obligate root symbionts, with arbuscular mycorrhizal fungi (AMF) playing particularly notable roles in root symbiosis [89]. These findings suggest that organic management may provide a more favorable environment for symbiotic interactions, enhancing nutrient uptake and tolerance to biotic stress in plants [90].

Furthermore, we identified distinct core microbiomes in the rice root endosphere (Figure 4A,B and Appendix A) and bulk soil (Figure 4C,D and Appendix A), suggesting the persistence of key bacterial taxa across different developmental stages and management types (Figure 4E,F). Regardless of external inputs, this stable core community likely plays a crucial role in maintaining essential ecosystem functions, highlighting the potential of microbiome management in sustainable agriculture.

### 4.3. Implications of Microbial Biomarkers for Rice Physiology and Paddy Ecology

The identification of stage-enriched microbial biomarkers has provided valuable insights into the functional roles of microbes under different management practices. During the tillering stage, methane-oxidizing bacteria (e.g., *Methylomonas*) were significantly enriched in the organic paddy (Figure 7A). According to a previous study [91], at this stage, microbial activity was considered to intensify, leading to increased methane production from organic matter decomposition. However, methanotrophs mitigate this effect by converting methane into carbon dioxide, thereby reducing greenhouse gas emissions. In contrast, in the conventional paddy, bacterial genera such as *Gaiella* and *Sphingomonas*, which are associated with the degradation of chemical compounds and microplastics in agricultural soils [60,61,62,63], were significantly enriched (Appendix A). The specific enrichment of these bacteria may result from the application of chemicals, including microbicides and plastic-coated chemical fertilizers, in the conventional paddy.

During the early ripening stage, roots in the organic paddy were enriched in symbiotic nitrogen-fixing bacteria such as *Bradyrhizobium* and *Azospirillum*, which may support spike and grain development by providing bioavailable nitrogen (Figure 7A). These symbiotic nitrogen-fixing bacteria effectively reduce the need for chemical nitrogen fertilizers by converting atmospheric nitrogen into a plant-absorbable form [92]. These findings align with previous studies [48,52], indicating that organic management practices can promote nutrient cycling while mitigating greenhouse gas emissions.

Moreover, the significant variation in microbial abundance across developmental stages suggests that the targeted introduction of beneficial microbes at specific growth stages could facilitate more sustainable and efficient management practices. These stage-enriched bacterial biomarkers in the rice root endosphere showed a significant positive correlation with the concentrations of macronutrients such as N, P, and K in leaves during the tillering stage in both organic and conventional paddy fields, but tended toward a negative correlation at the early ripening and maturing stages (Figure 7 and Appendix A).

These macronutrients described above are known to be highly mobile [93], and their concentrations in leaves likely decreased from the tillering stage to the early ripening and maturation stages due to remobilization from leaves to reproductive organs, such as panicles. In contrast, bacterial biomarkers in conventional paddy fields showed a significant correlation with low-mobility micronutrients, such as Fe, Ca, Mn, and Ni, in leaves during the early ripening and maturation stages (Figure 7).

Considering that changes in mineral nutrient concentrations in rice leaves across developmental stages showed both shared and distinct patterns between organic and conventional paddies (Appendix A), the leaf mineral nutrient profiles appear to reflect the influence of rice development and management practices. Therefore, the leaf mineral nutrient profile could serve as valuable information for improving the precision and reliability of microbial community comparisons in rice across different cultivation timings and management practices.

On the other hand, some stage-enriched bacterial root biomarkers, such as *Myxococcaceae_P3OB_42* and *Myxococcaceae_uncultured* in the early ripening stage, were identified (Figure 7). These biomarkers appeared to be unassociated with the profiles of mineral nutrient concentrations in rice leaves. The assembly of these bacterial biomarkers in rice roots may be correlated with biotic and abiotic environmental factors or agricultural management practices rather than the physiological state of the rice plants. Further studies will be required to investigate the factors driving the temporal assembly of these bacterial biomarkers.

### 4.4. Microbial Functions in Rice Paddies: Insights and Future Directions

Our results highlight the unique advantages of organic agricultural management in promoting rice health and ecological sustainability. Based on the findings in this study, we proposed a model illustrating the functional differences in root-associated microbial communities between organic and conventional paddy fields (Figure 8). In organic paddy fields, beneficial bacteria such as *Bradyrhizobium* and *Azospirillum* enhance rice growth through symbiotic nitrogen fixation, converting atmospheric nitrogen into ammonium and promoting nutrient absorption and root development. At the same time, diverse fungi and specific bacteria in organic paddy fields suppress pathogens through antagonistic interactions, enhancing rice disease resistance. In contrast, in conventional paddy fields, rice growth relies primarily on ammonium nitrogen from chemical fertilizers, with pesticides controlling pathogens to prevent disease. Moreover, these chemical applications in conventional paddy fields lead to the specific enrichment of bacteria involved in the degradation of chemical compounds and microplastics.

While this study provides valuable insights into the dynamic changes in microbial communities in the rice root endosphere, certain limitations must be acknowledged. To date, most research on the rice root microbiome has been conducted in greenhouse conditions with limited consideration for environmental factors such as climate change. Although this study was conducted under field conditions, our findings have yet to be validated in organic and conventional paddy fields across diverse geographic locations. Thus, further investigation is required to confirm the reproducibility of these conclusions. While we identified key microbial biomarkers in rice root and soil, our understanding of the functional roles of these microbial communities, especially fungal communities, remains incomplete. Multi-omics approaches, including metagenomics and metabolomics, are necessary to fully elucidate these specific roles.

From a broader perspective, effective management of the plant microbiome holds vast potential to sustainably enhance crop productivity and resilience, aligning with the global demand for increased food and biofuel production. However, successfully incorporating beneficial crop microbiomes into agricultural practices requires a deeper understanding of crop-microbiome interactions within modern agricultural systems. Future research should prioritize elucidating the molecular mechanisms through which microbial communities synergize with plant hosts to optimize nutrient cycling and enhance resilience to biotic and abiotic stresses. Additionally, long-term studies are crucial to assess the cumulative impacts of agricultural practices on microbial diversity and ecosystem services. Embedding microbial resource management into sustainable agricultural practices will be essential to address global challenges related to food security and environmental sustainability, ultimately supporting a more resilient agricultural future.

## 5. Conclusions

This study suggests that the agricultural management practices and rice developmental stages influence the dynamics of root-associated microbial communities. The organic paddy exhibited greater microbial dynamics, with symbiotic microorganisms enhancing nutrient cycling and supporting plant health, particularly during the early ripening stage. In contrast, the conventional paddy displayed a more stable microbial community structure, primarily characterized by its role in decomposing chemical compounds and microplastics in the soil. Furthermore, the composition and function of rice root-associated microbial communities in both paddy fields were influenced by soil physicochemical properties, such as ammonium nitrogen and iron, as well as by rice physiology, including leaf mineral macronutrient profiles. These results demonstrate how agricultural management practices and rice developmental stages influence microbial community composition and function, with significant implications for plant health and ecosystem sustainability. Additionally, these findings provide valuable scientific insights for optimizing agricultural practices, balancing productivity with environmental protection. They also highlight the importance of leveraging microbial resources to reduce chemical inputs, paving the way for the development of greener and more sustainable agricultural systems.

## Figures and Tables

**Figure 1 microorganisms-13-00041-f001:**
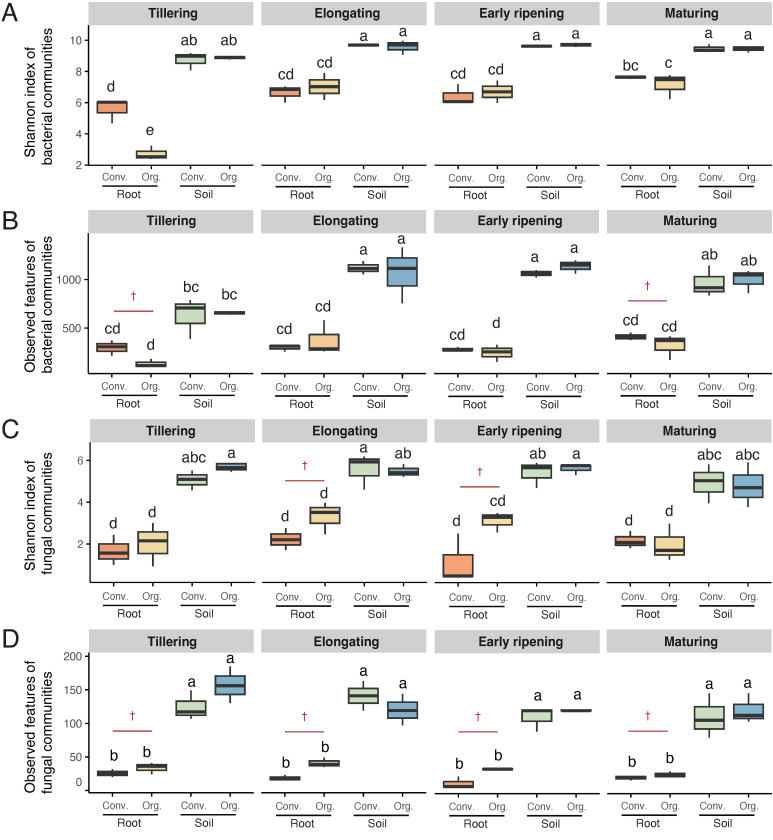
Temporal shifts in alpha-diversity of rice root and soil microbial communities across paddy types. (**A**–**D**) Shannon index and observed feature richness of bacterial (**A**,**B**) and fungal (**C**,**D**) communities in the root endosphere and bulk soil across four rice developmental stages: tillering, elongating, early ripening, and maturing. The box plots are colored as follows: brownish red for “Conv. Root”, beige for “Org. Root”, light-green for “Conv. Soil”, and blue for “Org. Soil”, “Conv.” and “Org.” represent conventional and organic paddies, respectively. Statistical significance was tested by one-way ANOVA followed by Tukey’s post hoc test. Different letters indicate statistically significant differences (*p* < 0.05). A dagger (†) indicates a large effect size (Cohen’s *d* > 0.8) between organic and conventional paddies where no statistically significant difference was detected.

**Figure 2 microorganisms-13-00041-f002:**
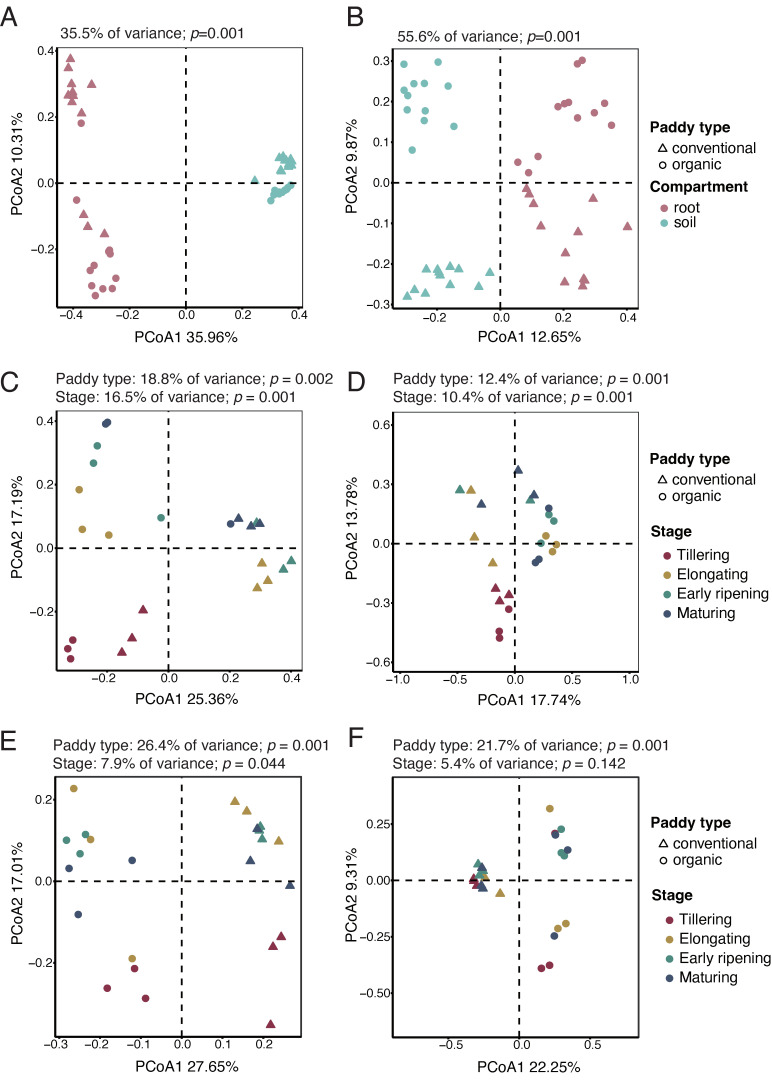
Temporal shifts in beta-diversity of rice root and soil microbial communities across paddy types. (**A**–**F**) Principal coordinate analysis (PCoA) based on Bray–Curtis distances illustrating the beta diversity of microbial communities across compartments (the root endosphere and bulk soil) and paddy types (organic and conventional). (**A**,**B**) PCoA plots of the overall bacterial (**A**) and fungal (**B**) communities, with root (red) and bulk soil (blue) samples under organic (circles) and conventional (triangles) paddies. (**C**,**D**) PCoA plots of bacterial (**C**) and fungal (**D**) communities in the root endosphere across four developmental stages and paddy types. (**E**,**F**) PCoA plots of bacterial (**E**) and fungal (**F**) communities in bulk soil across four developmental stages and paddy types. Significance values from PERMANOVA analysis, including R^2^ values and *p*-values, were annotated on the PCoA plots to highlight the degree of variation explained by compartment, paddy type, and developmental stage.

**Figure 3 microorganisms-13-00041-f003:**
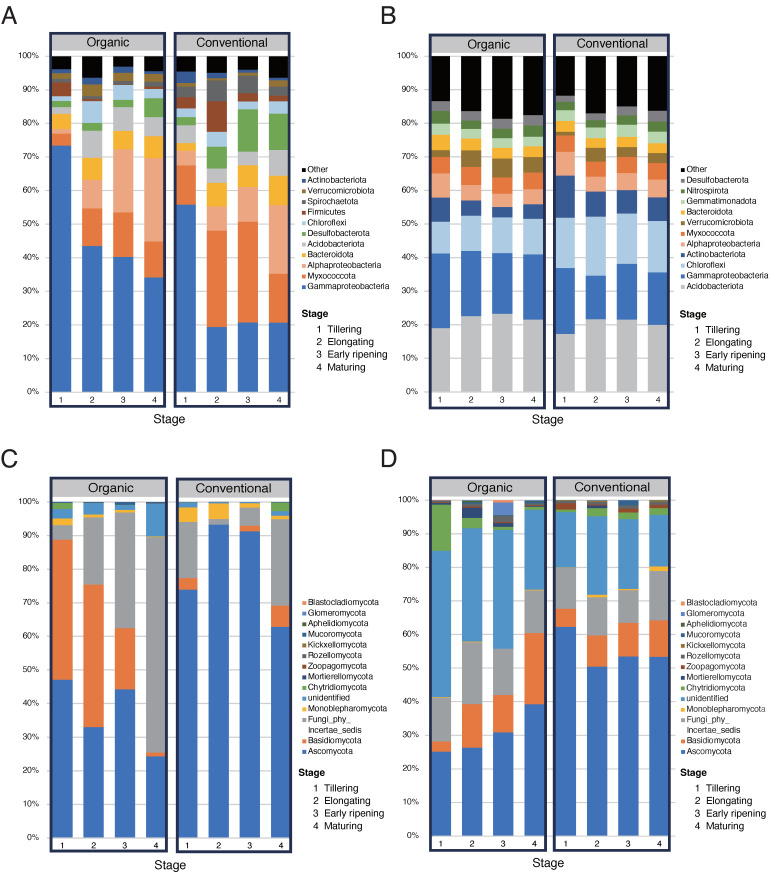
Temporal shifts in rice root and soil microbial community structure across paddy types. (**A**–**D**) Dominant microbial taxonomic groups in different compartments (the root endosphere and bulk soil) and paddy types (organic and conventional). (**A**,**B**) Bacterial community composition in the root endosphere (**A**) and bulk soil (**B**). (**C**,**D**) Fungal community composition in the root endosphere (**C**) and bulk soil (**D**). Stages 1, 2, 3, and 4 correspond to the tillering, elongating, early ripening, and maturing stages, respectively.

**Figure 4 microorganisms-13-00041-f004:**
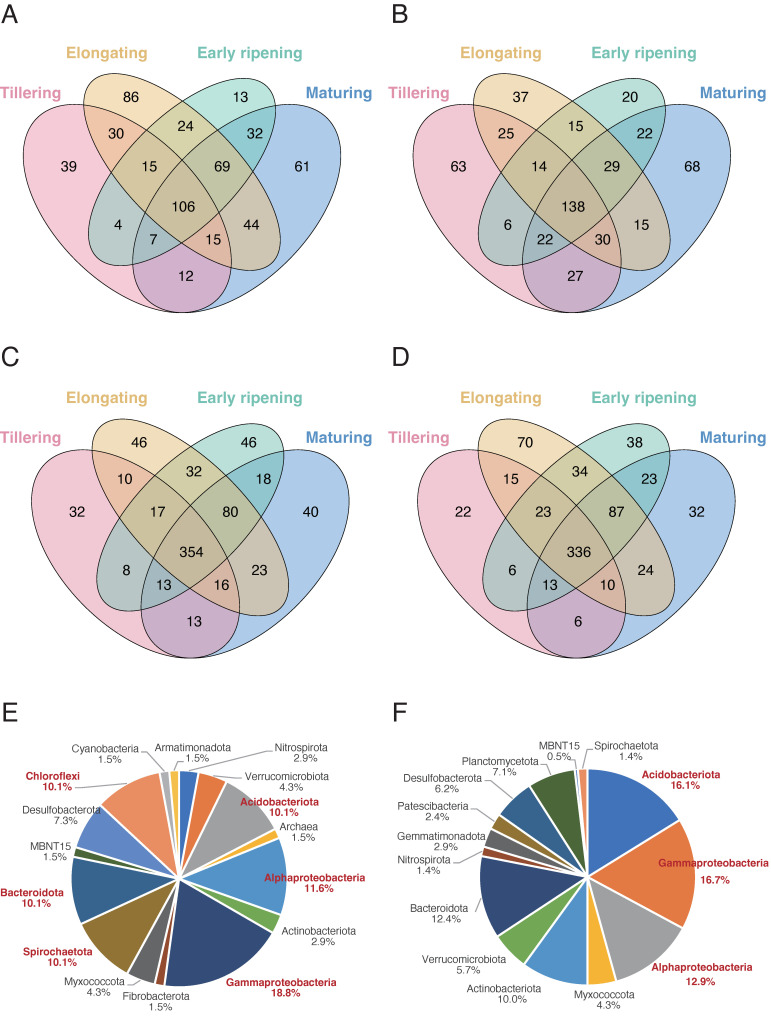
Core genera of rice root and soil bacterial communities. (**A**–**F**) Venn diagrams and pie charts of bacterial genera in the rice root endosphere and bulk soil across four developmental stages under organic and conventional paddies. (**A**,**B**) Venn diagrams showing the unique and shared genera in the root endosphere under organic (**A**) and conventional (**B**) paddies. (**C**,**D**) Venn diagrams showing the unique and shared genera in bulk soil under organic (**C**) and conventional (**D**) paddies. Each of the colored ovals represents the sampled stage. Values within intersections represent shared genera, while values outside intersections are unique to each stage. (**E**,**F**) Pie charts of the shared core bacterial microbiome, with 69 genera in the root endosphere (**E**) and 280 genera in bulk soil (**F**), summarizing the dominant phyla in the shared microbiomes. The top three phyla, ranked by proportion, are highlighted in red.

**Figure 5 microorganisms-13-00041-f005:**
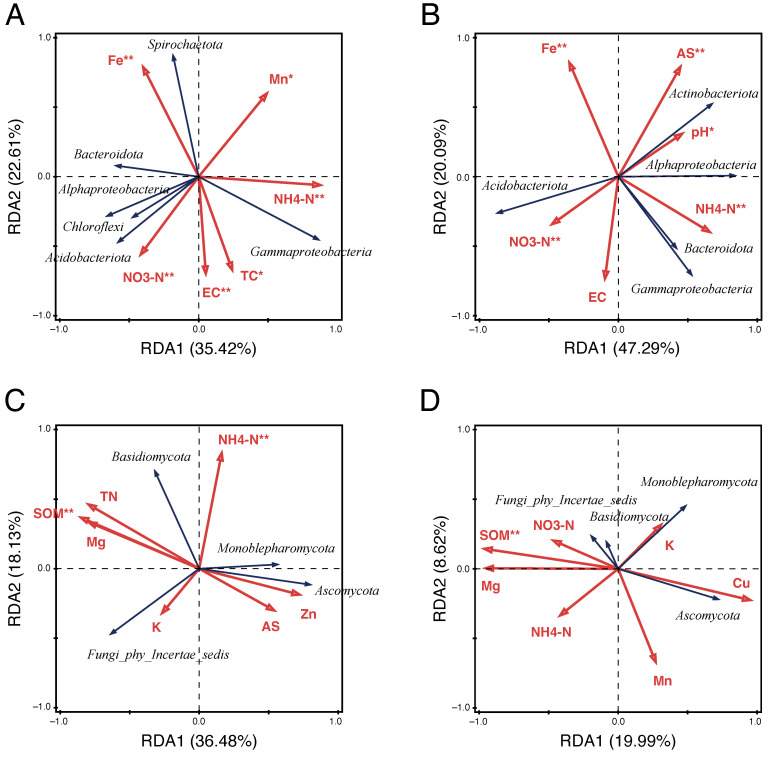
Correlation between dominant rice root and soil microbial communities and soil physicochemical properties. (**A**,**B**) Redundancy analysis (RDA) plots showing relationships between the dominant bacterial phyla in the root endosphere (**A**) and bulk soil (**B**) and soil physicochemical properties. (**C**,**D**) RDA plots showing relationships between the dominant fungal phyla in the root endosphere (**C**) and bulk soil (**D**) and soil physicochemical properties. Red and black arrows indicate soil physicochemical factors and dominant microbial phyla, respectively. Key soil physicochemical variables include ammonium nitrogen (NH₄⁺-N), total nitrogen (TN), total carbon (TC), nitrate nitrogen (NO_3_⁻-N), electrical conductivity (EC), soil organic matter (SOM), available silica (AS), pH, magnesium (Mg), zinc (Zn), manganese (Mn), copper (Cu), potassium (K), and iron (Fe). Arrow lengths denote the strength of influence, and the angle between soil physicochemical factors and microbial taxa indicates their relationship: acute angles represent positive correlations, while obtuse angles indicate negative correlations. The percentage of variance explained by each RDA axis is indicated along the axes. Asterisks indicate a significant correlation at *p* < 0.05, while double asterisks indicate a highly significant correlation at *p* < 0.01.

**Figure 6 microorganisms-13-00041-f006:**
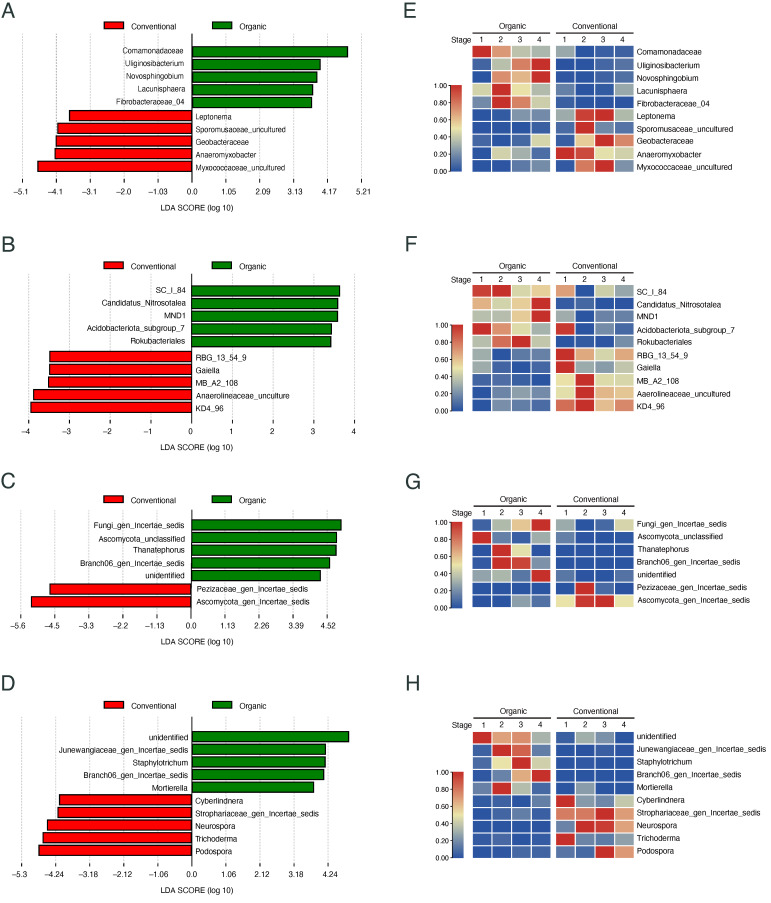
Differential rice root and soil microbial biomarkers between organic and conventional paddies. (**A**–**D**) Linear discriminant analysis (LDA) scores of bacterial and fungal biomarkers with LDA > 2 in the root endosphere (**A**,**C**) and bulk soil (**B**,**D**). Red bars indicate biomarkers enriched in the conventional paddy, while green bars represent biomarkers enriched in the organic paddy. (**E**–**H**) Heatmaps showing the relative abundance of biomarkers across four developmental stages in organic and conventional paddies, for the root endosphere (**E**,**G**) and bulk soil (**F**,**H**). The relative abundance of each genus was normalized to a maximum value of 1.00 for comparison. Stages 1, 2, 3, and 4 correspond to the tillering, elongating, early ripening, and maturing stages, respectively. Colors represent relative abundance, with red indicating higher abundance and blue indicating lower abundance. Detailed taxonomy information corresponding to microbial taxa in the microbial databases (the SILVA 138 database for bacteria and the UNITE database for fungi) is provided in Appendix A. The ’unidentified’ taxa represent sequences not confidently assigned to known species in the microbial reference databases. These taxa in different plots are independently derived and may not correspond to the same microbial groups.

**Figure 7 microorganisms-13-00041-f007:**
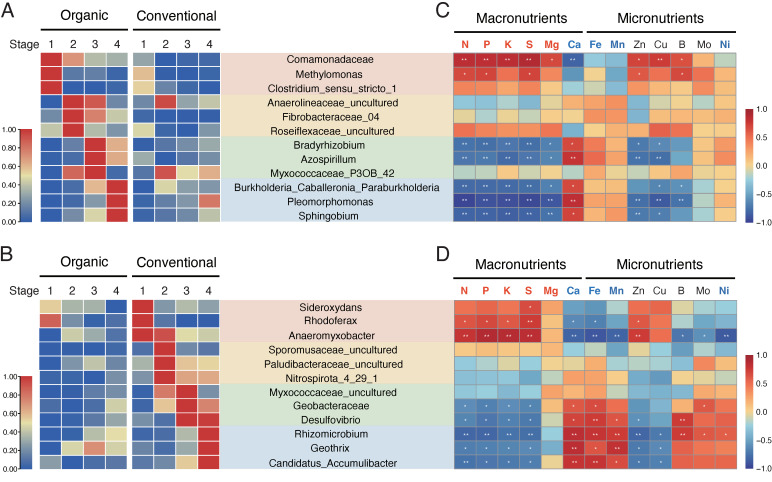
Developmental stage-enriched rice root bacterial biomarkers and their correlation with leaf mineral nutrient concentrations. (**A**,**B**) Heatmaps showing the relative abundance of bacterial biomarkers across each developmental stage in the rice root endosphere in organic (**A**) and conventional (**B**) paddies. The relative abundance of each genus was normalized to a maximum value of 1.00 for comparison. (**C**,**D**) Heatmaps showing the correlations between bacterial biomarkers in the root endosphere and concentrations of leaf mineral nutrients, including six macronutrients and seven micronutrients, in rice under organic (**C**) and conventional (**D**) paddies. The concentrations of each mineral nutrient were normalized such that the maximum deviation from the mean was set to 1.0 for comparison. Red and blue indicate highly mobile and low-mobility mineral nutrients, respectively. Significant correlations are marked by * (*p* < 0.05) and ** (*p* < 0.01). Stages 1, 2, 3, and 4 correspond to the tillering, elongating, early ripening, and maturing stages, respectively. The colors represent biomarkers enriched at different developmental stages: light-pink for tillering, light-beige for elongating, pastel-green for early ripening, and pale-blue for maturing. Detailed taxonomy information corresponding to microbial taxa in the SILVA 138 database is provided in Appendix A.

**Figure 8 microorganisms-13-00041-f008:**
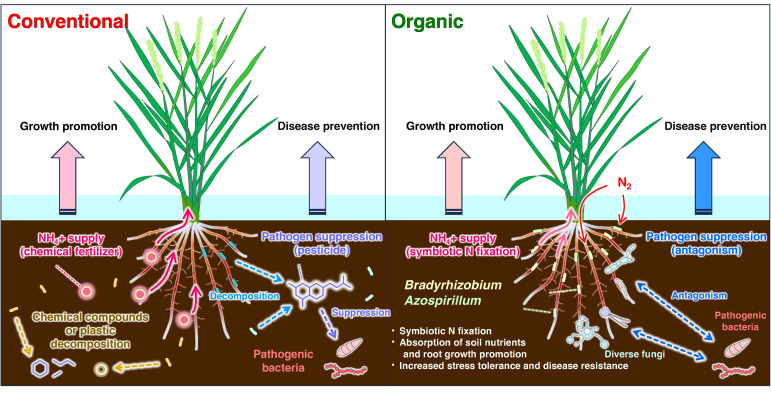
Proposed model for root-associated microbial functions in conventional and organic paddy fields. The diagram illustrates key differences in microbial processes between conventional (**left**) and organic (**right**) paddy fields. The white area represents the atmosphere. The blue area represents the water layer. The brown background represents the paddy soil environment. Dots of different colors represent bacteria with different functions, specifically, light-yellow and light-green dots represent *Bradyrhizobium* and *Azospirillum*, respectively. In conventional paddies, plant growth is promoted by chemical fertilizers that supply NH₄⁺, while disease prevention relies on pathogen suppression through microbicides. In organic paddies, rice growth is promoted via symbiotic nitrogen fixation by beneficial bacteria, such as *Bradyrhizobium* and *Azospirillum*, which convert atmospheric N_2_ into NH_4_⁺, enhancing nutrient absorption and root growth. Disease prevention is achieved through microbial antagonism, with diverse fungi and bacteria suppressing pathogens, resulting in improved stress tolerance and resistance.

## Data Availability

Sequence data were deposited into the NCBI Sequence Read Archive (SRA) database with the accession numbers PRJNA1177627 and PRJNA1180211.

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
