# Peer review of "A Comparison of Rice Root Microbial Dynamics in Organic and Conventional Paddy Fields"

_microorganisms, 2024, doi:10.3390/microorganisms13010041_

Round 1
Reviewer 1 Report
Comments and Suggestions for Authors
The article submitted by Masaru Fujimoto to Microorganisms is devoted to the study of the bacterial and fungal microbiomes of the rice rhizosphere grown under natural conditions, with mineral fertilizers
1) I suggest that the authors formulate the hypothesis that they tested during the study in the Introduction section; and in the Conclusion, describe whether it was proven
2) There are a number of issues that, in the reviewer's opinion, should be resolved prior publication. And will help make the presentation of the results more understandable for the reader
2.1) I believe that the Figure# captions can be removed from the figures
2.2.) Please, explain the exact meanings of Petzacaceae_gen_Incertae_sedis, Branch06_gen_Incertae_sedis and other taxons (P9X2b3D02, SBR1031, etc). This should be done throughout the manuscript, since such incomprehensible taxon designations are regularly encountered.
2.3) Which "unindentified" species are presented in Fig. 6? Are the sequences in all plots related to the same "unindentified" bacteria/fungi?
2.4) The same is related to the SC_I_84 and other abbreviations.
I believe that the article can be recommended for publication after these issues are eliminated
Author Response
Comments 1: I suggest that the authors formulate the hypothesis that they tested during the study in the Introduction section; and in the Conclusion, describe whether it was proven
Response 1: We have revised the Introduction section to explicitly state the hypothesis tested in this study. Additionally, we have updated the Conclusion section to describe whether the hypothesis was supported by the results. These revisions aim to provide a clearer framework for the study and improve the coherence of the manuscript. We hope these changes address your concerns.
2: There are a number of issues that, in the reviewer's opinion, should be resolved prior publication. And will help make the presentation of the results more understandable for the reader
Comments 2.1: I believe that the Figure# captions can be removed from the figures
Response 2.1: Removed.
Comments 2.2: Please, explain the exact meanings of Petzacaceae_gen_Incertae_sedis, Branch06_gen_Incertae_sedis and other taxons (P9X2b3D02, SBR1031, etc). This should be done throughout the manuscript, since such incomprehensible taxon designations are regularly encountered.
Response 2.2: We appreciate your valuable feedback. We have provided detailed taxonomy information of all taxa in the heatmap based on database annotations (the SILVA 138 database for bacteria and the UNITE database for fungi), as shown in Table S6, S7, S8, and S9. To ensure clarity and improve readability for readers, these updates have been incorporated into the main text and supplementary materials.
Comments 2.3: Which "unidentified" species are presented in Fig. 6? Are the sequences in all plots related to the same "unidentified" bacteria/fungi?
Response 2.3: The "unidentified" species presented in Figure 6 represent taxa that could not be confidently assigned to a known species in the respective databases (the SILVA 138 database and the UNITE database). These sequences were labeled as "unidentified" due to limitations in current reference databases or their ambiguous phylogenetic placement. We have clarified in the manuscript that the "unidentified" taxa in different plots may not necessarily refer to the same bacterial or fungal groups, as they are independently derived from the datasets corresponding to specific samples. This clarification has been added to the legend of Figure 6 and S5 for better understanding.
Comments 2.4: The same is related to the SC_I_84 and other abbreviations.
Response 2.4: Thank you for your comment! These abbreviations represent specific taxa or sequences obtained from microbial reference databases. To improve clarity, we have provided detailed taxonomy information in the supplementary materials as Table S6 and S8.
Comments 3: I believe that the article can be recommended for publication after these issues are eliminated.
Response 3: Thank you very much for your insightful comments and suggestions. We have carefully revised the manuscript according to your feedback. We believe these modifications significantly enhance the quality and clarity of our research. We sincerely appreciate your valuable input and hope that the revised version meets your expectations.
Reviewer 2 Report
Comments and Suggestions for Authors
The present manuscript studied the microbial c0ommunity in bulk and rhizosphere soil in two different system. The results are a little buit obvious, is large known that the compposition in organic and conventional system promote different genera and species of microorganisms as well as bulk and plant root. However some info0rmations sounds good. The manuscript is well write, and Figures are clear and is easier to understand.
Author Response
Comment: The present manuscript studied the microbial community in bulk and rhizosphere soil in two different system. The results are a little buit obvious, is large known that the composition in organic and conventional system promote different genera and species of microorganisms as well as bulk and plant root. However some informations sounds good. The manuscript is well written, and Figures are clear and is easier to understand.
Response: We sincerely thank you for your thorough review and valuable feedback. Although no specific revisions were suggested, we have made improvements to the manuscript based on comments from other reviewers. We greatly appreciate your time and effort in reviewing our work and ensuring the quality of the manuscript.
Reviewer 3 Report
Comments and Suggestions for Authors
The manuscript mainly evaluated the soil and root microbial community under different plant stage in organic and conventional paddy fields. Results showed that agriculture practice can shape microbial community. Ammonium nitrogen, iron, and soil organic matter were key drivers of microbial composition. The research topic is intriguing, the conclusions are supported by the results, the references are appropriate and the findings offer valuable suggestions for agricultural production. There are also some shortcomings that need to be addressed before publication. My main comments and suggestions are listed as follows.
1. Line 107, please provide more information about the plots (like the size of each plot in each field). Did the three individual plants combined into one sample? How many replicates for each treatment?
2. Lines 115-118, did the ‘remaining soil’ in line 115, was the same samples in line 118 ‘Soil samples’? If they represent the same sample, I don’t think it was necessary to repeat ‘collected from a depth of 5-10 cm’, which may cause misunderstanding.
3. Lines 183-285, the results presented in many places lack the evidence from statistical analysis. For example, in line 191 (with higher bacterial and fungal ……), I did not see the statistical evidence. I think either one-way ANOVA was used to test the effect of plant growth stages, soil/root, or organic/conventional on diversity respectively, or three-way ANOVAs was used to test these three effects and their interactive effects on microbial diversity. However, I just saw the effect of plant growth stage and organic/conventional which indicated by * or different letters above the box in Fig. 1.
4. Lines 222-227, 243, the same problems about statistical analysis were present here. I agree PCoA clearly showed that microbial community from different samples separated into different groups. However, statistical analysis like permanova or anosim were still necessary.
5. How did the core taxa was identified? Did it mean the most abundant, or unique taxa?
6. Lines 522-523, 694-695, it was weird to say the organic paddy increased fungal diversity because its effect on fungal diversity was non-significant.
7. Did the rice yield tested? did the organic field had higher rice yield than conventional field?
Author Response
Comments 1: Line 107, please provide more information about the plots (like the size of each plot in each field). Did the three individual plants combined into one sample? How many replicates for each treatment?
Response 1: We have added the information about the plots as requested in Section 2.1 of the Materials and methods. For sequencing, three individual replicate samples were collected, and the data were combined into one sample for data analysis, except for the PCoA analysis, where individual replicates were analyzed separately.
Comments 2: Lines 115-118, did the ‘remaining soil’ in line 115, was the same samples in line 118 ‘Soil samples’? If they represent the same sample, I don’t think it was necessary to repeat ‘collected from a depth of 5-10 cm’, which may cause misunderstanding.
Response 2: Revised.
Comments 3: Lines 183-285, the results presented in many places lack the evidence from statistical analysis. For example, in line 191 (with higher bacterial and fungal ……), I did not see the statistical evidence. I think either one-way ANOVA was used to test the effect of plant growth stages, soil/root, or organic/conventional on diversity respectively, or three-way ANOVAs was used to test these three effects and their interactive effects on microbial diversity. However, I just saw the effect of plant growth stage and organic/conventional which indicated by * or different letters above the box in Fig. 1.
Response 3: Thank you for your detailed and insightful comments. We appreciate your observation regarding the lack of explicit statistical evidence in certain parts of the Results section. To address this, we have clarified the statistical methods used and updated the text in Section 3.1 and 3.3 of the Results to ensure consistency with the data presented in Figure 1 and the newly added Figure S2. We have provided the results of the statistical analysis (p-values) to support the statement regarding higher bacterial and fungal diversity. Additionally, we have updated the numbering of supplemental figures throughout the manuscript to reflect the inclusion of the new Figure S2.
We confirm that one-way ANOVA was used for Figure 1, Figure S1, and Figure S2A, and while two-way ANOVA was applied for Figure S2B to examine the effects of developmental stages, soil/root compartments, and organic versus conventional management on microbial diversity individually. Where applicable, Tukey’s HSD post-hoc test was performed for pairwise comparisons, and these results are now explicitly referenced in the manuscript.
In addition, as per your suggestion, we have included the results of a three-way ANOVA to show the interactive effects of developmental stage, compartment, and paddy type on microbial diversity in the Supplementary Materials (Table S2, S3, S4 and S5) and these have been referenced in Section 3.1 of the Results. Furthermore, to complement these changes, we have updated Section 4.1 and 4.2 of the Discussion to incorporate the statistical analyses and discuss their implications for our findings.
We hope these updates address your concerns and provide greater clarity in the presentation of our results. Thank you again for your valuable suggestions.
Comments 4: Lines 222-227, 243, the same problems about statistical analysis were present here. I agree PCoA clearly showed that microbial community from different samples separated into different groups. However, statistical analysis like permanova or anosim were still necessary.
Response 4: Thank you for your insightful comment regarding the need for statistical validation of the PCoA results. We agree that additional statistical analysis is essential to support the observed separation of microbial communities among different groups.
In response to your comment, we have conducted PERMANOVA analysis to statistically validate the differences in microbial community composition among the groups shown in the PCoA. The PERMANOVA results, including p-values and R² values, have been added to Figure 2, Section 3.2 of the Results, and Section 4.1 of the Discussion. These updates aim to provide robust statistical support for the PCoA findings and strengthen the conclusions of our study. We sincerely thank you for your constructive feedback, which has helped improve the rigor of our manuscript.
Comments 5: How did the core taxa was identified? Did it mean the most abundant, or unique taxa?
Response 5: The core taxa were identified as the genera that are consistently present across all developmental stages of rice. This reflects the common shared microbial community that persists throughout the growth cycle, regardless of abundance or uniqueness. The identification of core taxa highlights the stable and essential members of the rice microbiome, which may play fundamental roles in supporting plant development and resilience.
Comments 6: Lines 522-523, 694-695, it was weird to say the organic paddy increased fungal diversity because its effect on fungal diversity was non-significant.
Response 6: Thank you for your comment highlighting the need to evaluate the statistical significance of the observed trends. While the one-way ANOVA results indicated that the differences in fungal diversity between organic and conventional paddies were not statistically significant (p > 0.05), we conducted an additional effect size analysis to assess the practical significance of these differences. Effect size analysis revealed large effect sizes (Cohen’s d > 0.8), indicating that the substantial differences in fungal diversity among groups are meaningful despite the lack of statistical significance. This discrepancy is likely due to the small sample size (n = 3), which may have limited the statistical power of ANOVA and Tukey HSD tests. Tukey HSD is particularly conservative under such conditions, which can further obscure the detection of differences even when group differences are relatively large. To provide a more comprehensive interpretation of the results, we have incorporated the effect size analysis into Figure 1C and 1D of the revised manuscript and discussed its implications in Section 3.1 of the Results and Section 4.1 of the Discussion.
Comments 7: Did the rice yield tested? did the organic field had higher rice yield than conventional field?
Response 7: We have added new data showing the shoot and panicle dry weights of rice plants at the maturing stage as Figure S7, along with the corresponding descriptions in Section 4.1 of the Discussion. The results showed no significant differences between the organic and conventional fields.
Thank you very much for your insightful comments and suggestions. We have carefully revised the manuscript according to your feedback. We believe these modifications significantly enhance the quality and clarity of our research. We sincerely appreciate your valuable input and hope that the revised version meets your expectations.
Round 2
Reviewer 3 Report
Comments and Suggestions for Authors
The author has made corresponding revisions based on my suggestions. I think the work is ready for publication.
Author Response
Response: Thank you very much for your thorough review and constructive suggestions. We are glad to hear that the revisions have addressed your concerns and that you consider the work ready for publication. Your feedback has been invaluable in improving the manuscript, and I greatly appreciate your time and effort in reviewing it.